# The Impact of PSMA-PET on Oncologic Control in Prostate Cancer Patients Who Experienced PSA Persistence or Recurrence

**DOI:** 10.3390/cancers15010247

**Published:** 2022-12-30

**Authors:** Lorenzo Bianchi, Francesco Ceci, Francesco Costa, Eleonora Balestrazzi, Matteo Droghetti, Pietro Piazza, Alessandro Pissavini, Riccardo Mei, Andrea Farolfi, Paolo Castellucci, Stefano Puliatti, Alessandro Larcher, Giorgio Gandaglia, Daniele Robesti, Alexandre Mottrie, Alberto Briganti, Alessio Giuseppe Morganti, Stefano Fanti, Francesco Montorsi, Riccardo Schiavina, Eugenio Brunocilla

**Affiliations:** 1Division of Urology, IRCCS Azienda Ospedaliero-University of Bologna, 40138 Bologna, Italy; 2University of Bologna, 40126 Bologna, Italy; 3Division of Nuclear Medicine, IEO European Institute of Oncology IRCCS, 20141 Milan, Italy; 4Department of Oncology and Hemato-Oncology, University of Milan, 20122 Milan, Italy; 5Nuclear Medicine, IRCCS Azienda Ospedaliero-University of Bologna, 40138 Bologna, Italy; 6Department of Urology, University of Modena and Reggio Emilia, 41122 Modena, Italy; 7Department of Urology, Onze-Lieve-Vrouwziekenhuis, 9300 Aalst, Belgium; 8ORSI Academy, 9300 Melle, Belgium; 9Unit of Urology, Division of Experimental Oncology, Urological Research Institute, IRCCS San Raffaele Scientific Institute, Vita-Salute San Raffaele University, 20132 Milan, Italy; 10Radiation Oncology, IRCCS Azienda Ospedaliero, University of Bologna, 40138 Bologna, Italy

**Keywords:** PSMA-PET, hormone sensitive prostate cancer, recurrent prostate cancer, PSMA-guided salvage treatment, survival

## Abstract

**Simple Summary:**

PSMA-PET is currently recommended to restage PCa and to guide salvage treatments. We aim to evaluate the oncologic outcomes of patients with recurrent PCa who received PSMA-PET. PSMA-PET may be a prognostic tool in BCR patients after PR. In recurrent PCa patients who never received previous salvage therapies, men with positive PSMA-PET had similar oncologic outcomes compared to those with negative PSMA-PET. PCa patients who already had previous salvage therapies with positive PSMA-PET experienced worse oncologic outcomes compared to those with negative PSMA-PET. In a PSMA-PET positive population no significant differences were found in terms of progression and metastasis between patients with oligometastatic vs. polimetastatic disease and local/N1 vs. M1 at PSMA-PET.

**Abstract:**

Background: Prostate Specific Membrane Antigen-Positron Emission Tomography (PSMA-PET) is currently recommended to restage prostate cancer (PCa) and to guide the delivery of salvage treatments. We aim to evaluate the oncologic outcomes of patients with recurrent PCa who received PSMA-PET. Methods: 324 hormone-sensitive PCa with PSA relapse after radical prostatectomy who underwent PSMA-PET in three high-volume European Centres. Patients have been stratified as pre-salvage who never received salvage treatments (*n* = 134), and post-salvage, including patients who received previous salvage therapies (*n* = 190). Patients with oligorecurrent (≤3 lesions), PSMA-positive disease underwent PSMA-directed treatments: salvage radiotherapy (sRT) or Metastases-directed therapy (MDT). Patients with polirecurrent (>3 lesions) PSMA-positive disease were treated with systemic therapy. Patients with negative PSMA-PET were treated with sRT or systemic therapies or observation. The primary outcome of the study was Progression-free survival (PFS). Secondary outcomes were: Metastases-free survival (MFS) and Castration Resistant Pca free survival (CRPC-FS). Results: median follow up was 23 months. In the pre-salvage setting, the PFS, MFS and CRPC-FS estimates at 3 years were 66.2% vs. 38.9%, 95.2% vs. 73.7% and 94.9% vs. 93.1% in patients with negative vs. positive PSMA-PET, respectively (all *p* ≥ 0.2). In the post-salvage setting, the PFS, MFS and CRPC-FS estimates at 3 years were 59.5% vs. 29.1%, 92.7% vs. 65.1% and 98.8% vs. 88.8% in patients with negative vs. positive PSMA-PET, respectively (all *p* ≤ 0.01). At multivariable analyses, a positive PSMA-PET was an independent predictor of progression (HR = 2.15) and metastatic disease (HR 2.37; all *p* ≤ 0.03). Conclusion: PSMA-PET in recurrent PCa detects the site of recurrence guiding salvage treatments and has a prognostic role in patients who received previous salvage treatments.

## 1. Introduction

Standard management for prostate cancer (PCa) patients with biochemical persistence (BPC) or recurrence (BCR) after radical treatments historically relied on salvage radiotherapy (RT) and/or hormonal therapy. The introduction of new generation imaging significantly influenced the clinical management of recurrent PCa patients [1]. Modern imaging led to better identification of patients with oligometastatic disease. Accordingly, there is increasing interest in metastasis-directed therapy (MDT) [2]. The rationale of MDT is to treat all active PCa metastases to prevent further metastatic spread and improve survival [3]. Two prospective phase II trials demonstrated the safety of MDT approaches in these settings and its efficacy in improving androgen deprivation therapy ADT-free survival [2,4]. The selection of patients eligible for MDT should rely on an imaging that accurately defines true oligo-recurrent diseases and on prognostic parameters to predict which patients are most likely to respond to ablative treatments [4].

Prostate Specific Membrane Antigen-Positron Emissions Tomography (PSMA-PET) represents the most accurate novel imaging procedure to restage PCa [5], due to its high diagnostic accuracy to correctly detect and localize PCa lesions [6] even in the early stage of disease, when the tumor burden and prostate specific antigen (PSA) levels are low [7]. Moreover, the use of PSMA-PET in restaging PCa patients lead to a change in clinical management in approximately 50% of cases [8,9,10], despite the fact that evidence of a survival benefit of treatment changes are scarce. However, the likelihood of positivity for PSMA-PET is influenced by several parameters, and different prediction models have been proposed to select patients for PSMA-PET imaging [11,12,13]. PSMA-PET could be the optimal tool to select the best candidates for MDT. However, data derived from phase III trials enrolling large cohorts of patients and powered for efficacy have yet to be completed [14]. Finally, it remains unclear whether PSMA-PET is a prognostic parameter associated with patient survival. Thus, the aim of this analysis was to evaluate the oncologic outcomes of patients who received PSMA-PET to stage the disease during Pca recurrence.

## 2. Materials and Methods

### 2.1. Study Population

The cohort of patients included in this study was enrolled through an open label, multicenter, retrospective analysis in three tertiary high-volume European centers (IRCCS Sant’ Orsola-Malpighi in Bologna, IRCCS San Raffaele in Milan, and the OLV Hospital in Aalst). In all centers, patients were enrolled in accordance with Institutional Review Board (IRB) and ethical committee approval and signed an informed consent form (ICF) as per local requirements. Clinical records of PCa patients who performed RP between January 1998 and January 2021 and PSMA-PET from January 2016 and February 2021 were analyzed. The inclusion criteria were: (1) proven hormone-sensitive PCa; (2) confirmed BCP or BCR according to EAU guidelines [1]; (3) a PSMA-PET scan performed after PSA relapse; and (4) patients who did not receive androgen deprivation therapy (ADT) for at least 6 months prior to PSMA-PET scan. Three-hundred and fifty-one (*n* = 351) individuals met the inclusion criteria. Patients (*n* = 7) who received previous chemotherapy or androgen receptor targeted agents (ARTA) and Castration resistant Prostate Cancer (CRPC) patients (*n* = 20) at the time of PSMA-PET were excluded. Three-hundred and twenty-four (*n* = 324) patients fully met the inclusion/exclusion criteria and were considered for primary end-point analysis. Two sub-populations have also been identified, and patients have been stratified as pre-salvage setting who never received salvage treatments (overall, *n* = 134), including patients with BCP (*n* = 50) or first BCR after RP (*n*= 84) and post-salvage setting, including patients with BCR who performed PSMA-PET for PSA relapse after previous salvage therapies after RP (*n* = 190).

### 2.2. PSMA-PET Procedure and Interpretation Criteria

In all centres, 68Ga-PSMA-11 was synthesized according to good manufacturing practice (GMP) and in accordance with international procedural guidelines [15,16]. A mean dose of 1.8–2.2 MBq/Kg body weight of 68Ga-PSMA-11 was administered intravenously. 68Ga-PSMA-11 PET/Computed Tomography (CT) was performed with a standard technique, and in accordance with international procedural guidelines [16]. All studies were performed using dedicated PET state-of-the-art scanners. All PSMA-PET images were locally reviewed prior to data sharing, independently by two experienced nuclear medicine physicians and according to international reporting guidelines [15,17].

### 2.3. Patients’ Management and Treatments

PSMA-PET findings have been interpreted according to international procedural guidelines [17]. In brief, PSMA-PET were considered negative (no PCa lesion) vs. positive (presence of suspected PCa lesions in any sites). Patients with positive PSMA-PET were stratified according to anatomic sites (Local and/or N1 vs. any M) and number of lesions (i.e., oligorecurrent [≤3 lesions] vs. polirecurrent Pca [>3 lesions]). In each center, therapies after PSMA-PET have been administered according to international urologic guidelines [1] and decisions on treatment management were taken by a multidisciplinary tumor board and considering patient preference. In brief, patients with oligorecurrent PSMA-PET positive disease underwent PSMA-guided salvage treatment that consisted of either salvage prostate bed RT/whole pelvis RT (for local recurrence) or MDT (including sLND for suspicious pelvic lymph nodes [N1] and stereotactic body RT [SBRT] for suspicious pelvic [N1] or extra-pelvic lymph nodes [M1a] and/or skeletal lesions [M1b]) according to relapse pattern. In a vast majority of cases, MDT was targeted to PSMA-PET positive lesions with no further confirmation by conventional imaging. ADT was allowed as adjuvant treatment according to international procedural guidelines [1], multidisciplinary tumor board decisions and patients preferences. Patients with polirecurrent PSMA-PET positive disease have been treated with systemic therapy (including ADT or a combination of ADT with chemotherapy or ARTA). Patients with negative PSMA-PET were treated with best clinical practice (including salvage prostate bed RT/whole pelvis RT or systemic therapies or observation), according to patients’ clinical status (including eventual comorbidities), previous treatments, and multidisciplinary tumor board decisions.

### 2.4. Outcomes Measurements

The primary outcome of the study was Progression-free survival (PFS), defined as the time in months between the date of PSMA-PET and the date of progression or last follow-up. Progression was defined as one of the following conditions: PSA progression after therapy, radiological progression defined as the appearance of new PCa localization(s) at any imaging procedure performed during follow-up according to best standard of care (including bone scan, contrast-enhanced CT, whole-body MRI, PSMA or choline or fluciclovine-PET), and death due to any cause. Secondary outcomes were: Metastases-free survival (MFS), defined as the appearance of new PCa metastases (i.e., M1 disease) at any imaging procedure performed during follow-up according to best standard of care (including bone scan, contrast-enhanced CT, whole- body MRI, PSMA or choline or fluciclovine-PET), and CRPC free survival (CRPC-FS), defined as the occurrence of CRPC1 (both metastatic and non-metastatic) during follow-up.

### 2.5. Statistical Analyses

Statistical analyses firstly consisted of descriptive statistics of the overall population and stratifying the population according to PSMA-PET result (negative vs. positive) in each subpopulation (namely, pre-salvage and post-salvage setting of recurrence). Chi-squared and Mann Whitney tests were used to compare proportions and medians between the two groups, respectively. The Kaplan-Meier method was used to assess PFS, MFS and CRPC-FS estimates at 3 years follow-up in the pre-salvage setting and post-salvage setting population separately after stratifying according to PSMA-PET results (negative vs. positive). In patients with positive PSMA-PET, PFS and MFS estimates at 3 years were evaluated in the pre-salvage setting and post-salvage setting population separately after stratifying patients according to according to anatomic sites (Local and/or N1 vs. any M) and number of lesions (i.e., oligorecurrent vs. polirecurrent Pca), as compared by the log-rank test. Third, multivariable Cox regression models were performed to identify independent predictors of PFS and MFS. Covariates were age, pathologic stage, pathologic ISUP group, pathologic N status, clinical setting (pre-salvage and post-salvage subpopulation), PSA at PSMA-PET scan and PSMA-PET results (positive vs. negative). All statistical tests were performed with R 4.0.3 (R Foundation for Statistical Computing, Vienna, Austria) with a two-sided significance level set at *p* < 0.05.

## 3. Results

The median PSA at PSMA-PET was 0.5 ng/mL. Overall, 134 (41.4%) and 190 (58.6%) men underwent PSMA-PET at pre-salvage and post-salvage setting, respectively (Table 1).

Table 2 showed the treatments performed after PSMA-PET and the oncologic outcomes after PSMA-PET. The median follow-up after PSMA-PET was 23 months (IQR:10–34; Table 2).

### 3.1. Pre-Salvage Setting (n = 134)

In the pre-salvage setting, the PSMA-PET positivity rate was 52% (69 out of 134 patients). Patients with positive PSMA-PET had significant higher pathologic T stage, pN1 status, pathologic ISUP group and median PSA level at PSMA-PET compared to patients with negative ones (all *p* ≤ 0.03; Table 1). Appendix A shows baseline characteristics in patients with positive PSMA-PET (*n* = 193) stratified according to the localization of positive lesions (local and/or N1 vs. M1). Appendix A shows baseline characteristics in patients with positive PSMA-PET (*n* = 193) stratified according to the stage of disease (oligorecurrent vs. polirecurrent).

PFS, MFS and CRPC-FS estimates at 3 years were 66.2% vs. 38.9% (*p* = 0.3; Figure 1a), 95.2% vs. 73.7% (*p* = 0.2; Figure 2a) and 94.9% vs. 93.1% (*p* = 0.3; Figure 3a) in patients with negative vs. positive PSMA-PET, respectively.

### 3.2. Pre-Salvage Setting (n = 190)

In the post-salvage setting, the PSMA-PET positivity rate was 65% (124 out of 190 patients). Patients with positive PSMA-PET had significantly lower age (*p* = 0.02), but no significant differences were found concerning pathologic characteristics compared to patients with negative ones (Table 1).

PFS, MFS and CRPC-FS estimates at 3 years were 59.5% vs. 29.1% (*p* < 0.001; Figure 1b), 92.7% vs. 65.1% (*p* < 0.0004; Figure 2b), and 98.8% vs. 88.8% (*p* = 0.01; Figure 3b) in patients with negative vs. positive PSMA-PET, respectively. Considering patients with positive PSMA-PET, no significant differences were found concerning PFS and MFS estimates at 3 years after stratifying according to disease localization at PET scan (namely, local and/or N1 vs. M1 disease) and according to the number of positive lesions at PSMA-PET (namely, oligorecurrent [≤3 lesions] vs. polirecurrent [>3 lesions]) both in the pre-salvage and post-salvage setting (all *p* ≥ 0.2; Appendix A). In the multivariable Cox regression analysis, positive PSMA-PET (HR = 2.15; 95% CI: 1.42–3.25), age (HR = 0.97; 95% CI: 0.95–0.99), pT3b-pT4 (HR = 1.84; 95% CI: 1.10–3.09) and ADT at salvage treatment (HR = 0.58; 95% CI 0.97–1.02) were independent predictors of progression (all *p* ≤ 0.03). Positive PSMA-PET (HR 2.37; 95% CI 1.60–3.50), pT3b-pT4 (HR = 2.03; 95% CI: 1.26–3.26) and ADT at salvage treatment (HR = 0.54; 95% CI 0.37–0.78) were independent predictors of metastatic disease (all *p* ≤ 0.003; Table 3).

## 4. Discussion

The introduction of PSMA-PET in the management of recurrent PCa generated patients’ migration to a metastatic disease in earlier stages. Oligorecurrent and oligometastatic patients can be treated with MDT [2,18], thus increasing the interest in PSMA-guided therapies. However, there is a lack of evidence regarding the long-term benefit of this approach on oncological outcomes [19]. In the current multicentric study, we performed a survival analysis in hormone-sensitive PCa patients with PSA recurrence after RP who underwent PSMA-PET, and salvage treatments guided by PSMA imaging, evaluating the potential prognostic role of PSMA-PET. In the recurrent setting, PSMA-PET represents a game-changing procedure, identifying those patients at higher-risk who should be treated with a personalized approach in cases of oligorecurrent disease or the combination of systemic therapy in cases of polimetastatic disease [20].

Patients with low PSA levels (a median of 0.5 ng/mL), suitable for potentially curative salvage treatments were considered, and the PSMA-PET positivity rate significantly differed in salvage therapy naïve patients (36%) and in patients who received PSMA-PET scans for PSA recurrence after previous salvage therapy (64%). The setting of recurrence reflects the natural history of PCa and may influence both the results of the PSMA-PET [12] treatments available and oncologic outcomes. In the pre-salvage population (including BPC and patients with first BCR after RP), a positive PSMA-PET does not represent a prognostic tool, since patients with positive PSMA-PET had comparable PFS, MFS and CRPC-FS at 3 years compared to men with negative imaging. In this cohort, a positive PSMA-PET identifies men with “macroscopic” recurrent disease (oligorecurrent in most cases) suitable for aggressive curative treatments including conventional salvage RT, different type of MDT and systemic therapies. The condition of proven BCR and negative PSMA-PET might be related to the presence of micro-metastatic disease not detectable by molecular imaging, which may achieve optimal oncologic outcomes with conventional salvage treatments (i.e., conventional salvage RT), or less aggressive recurrent disease even suitable for observation in selected cases, due to the lower risk of adverse oncologic outcomes. The sub-optimal PSMA-PET accuracy in the early stages of the disease, especially in the case of indolent recurrence, might be explained by the limited resolution of current PET scanners, as only lesions greater than 3–4 mm can be detected adequately. Moreover, a low PSA level may reflect a negative PSMA-PET and could be a potential bias for PSMA-PET diagnostic performance. In addition, up to 5% of all PCa do not harbor significant PSMA expression [12]. In the post-salvage population (including men who already received previous salvage treatments), positive PSMA-PET was a prognostic parameter as PET positive patients had significantly lower PFS, MFS and CRPC-FS at 3 years compared to men with negative ones (29.1% vs. 59.5%, 65.1% vs. 92.7% and 88.8% vs. 98.4%, all *p* ≤ 0.001).

Accordingly, a less aggressive approach (including observation or ADT) could be offered in case of low-risk disease [21] (low PSA, long PSA doubling time, low ISUP, and negative PSMA-PET) considering age and previous oncological treatments, since patients with negative PSMA-PET had a lower risk of progression. However, prospective data are needed to confirm this hypothesis.

On the contrary, a positive PSMA-PET identifies a population at higher-risk of a less favorable prognosis. In this setting, the earlier identification of metastatic patients, compared to conventional imaging, lead to the anticipation of specific treatments in an early stage, despite the fact that clinical management is influenced by previous salvage treatments performed. Patients with positive PSMA-PET are a heterogenous group with different prognoses. Indeed, the early identification of the oligometastatic stage may identify patients suitable for PSMA-guided MDT [22]. However, despite the adoption of a modern approach to positive PSMA-PET lesions including MDT for oligometastatic and novel multidrug approaches for polimetastatic disease, the prognosis of men with positive PSMA-PET is poorer and the risk of metastatic progression and evolution to CRPC status is still high. Further efforts should evaluate different PET parameters to identify patients for MDT who may achieve the best oncologic benefit and the ideal combination approach for MDT. Indeed, results from the PEACE V-STORM trial [23] would definitely assess whenever the treatment of all positive PET lesions by MDT combined with whole pelvis RT would be beneficial in nodal oligometastatic Pca patients compared to an MDT alone approach.

In patients with a positive PSMA-PET, no significant differences were found concerning PFS and MFS after stratifying the population considering the number of positive lesions (oligometastatic vs. polimetastatic disease) or the site of relapse (local and N1 vs. M1 disease) both in pre-salvage and post-salvage settings. The potential candidates for PSMA-guided MDT should be men with oligometastatic disease (≤3/≤5 lesions) and N1 or M1a-b. However, even in highly selected patients with PSMA-PET for ablative SBRT to nodal or bone metastases, 25% of cases would experience immediate PSA progression [22]. This could be due to the presence of residual micro-metastases that remain undetectable. However, the consolidation of macroscopic metastases may remove or significantly affect signals that promote the development of remaining micro-metastases as suggested by the lower risk of new metastases at 6 months in men treated with MDT of all detectable lesions by PSMA-PET4.

Finally, at multivariate Cox regression, a positive PSMA-PET was found to be an independent predictor of progression (HR 2.15) and metastases (HR: 2.37). These findings suggest that when PSMA-PET is performed to restage PCa patients, the results of PSMA-PET is a prognostic parameter for oncologic outcomes, helping the treating physicians to guide salvage therapies, but also to adopt a more conservative approach in the case of a negative scan [24].

PSMA-PET can be integrated together with further novel biomarkers, including ctDNA, exosomes and genomic panels to improve the selection of candidates for novel personalized therapy.

## 5. Limitation

Despite several strengths, our study is not devoid of limitations. First, the retrospective design of the study may have influenced the selection process of our cohort. However, these data have been derived in each center by prospective studies in consecutive patients. Second, even if a central review was not performed, all PSMA-PET images were evaluated with a local review by PSMA-PET–experienced nuclear medicine physicians according to international reporting procedural guidelines. Third, the short follow-up time after PSMA limited further consideration of long-terms outcomes. Fourth, the histologic validation of positive findings was not feasible in all cases due to ethical and practical reasons, and thus the presence of false positive findings cannot be excluded. However, in registry trials [25], PSMA-PET demonstrated optimal positive predictive value. Finally, no direct comparison with conventional imaging was performed in terms of treatment change and prognostic effect, since most patients were only investigated with PSMA-PET according to recent EAU recommendations.

## 6. Conclusions

In recurrent PCa patients after surgery, PSMA-PET could be used to select specific and personalized treatments and should be considered as a prognostic parameter in patients who received previous salvage treatments, since a positive scan is associated with shorter PFS, MFS, and CRPC-FS. In salvage treatment naïve patients, a positive PSMA-PET had similar oncologic outcomes compared to negative ones. Patients with positive PSMA-PET are a group at higher-risk, and thus justify the adoption of a modern approach, including MDT for oligometastatic disease and novel multidrug approaches for polimetastatic disease. Finally, the specific impact of image-guided therapy on the overall survival of patients affected by BCR remains to be clarified.

## Figures and Tables

**Figure 1 cancers-15-00247-f001:**
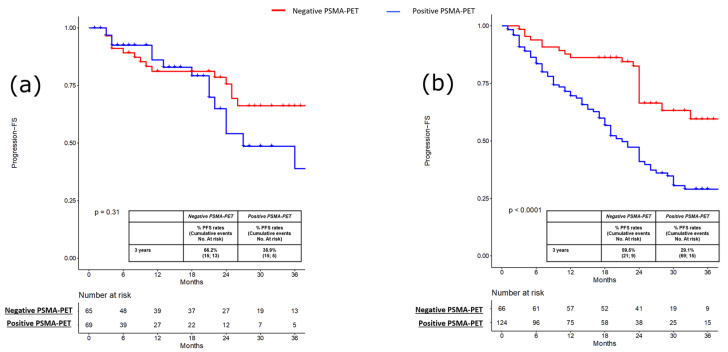
(**a**) Kaplan-Meier curve depicting Progression Free Survival (PFS) rates in the pre-salvage setting (*n* = 134) according to PSMA-PET results (namely, positive vs. negative); (**b**) Kaplan-Meier curve depicting Progression Free Survival (PFS) rates in the post-salvage setting (*n* = 190) according to PSMA-PET results (namely, positive vs. negative).

**Figure 2 cancers-15-00247-f002:**
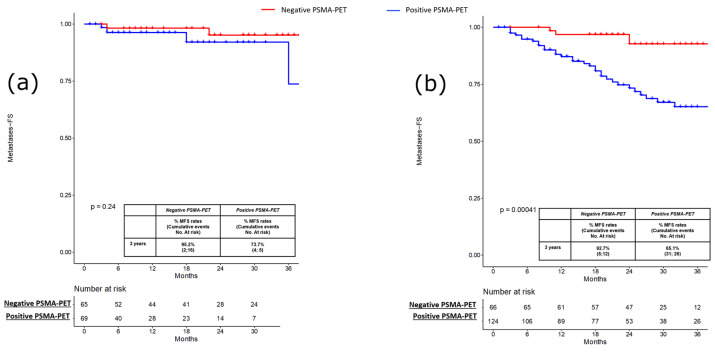
(**a**) Kaplan-Meier curve depicting Metastases Free Survival (MFS) rates in the pre-salvage setting (*n* = 134) according to PSMA-PET results (namely, positive vs. negative); (**b**) Kaplan-Meier curve depicting Metastases Free Survival (MFS) rates in the post-salvage setting (*n* = 190) according to PSMA-PET results (namely, positive vs. negative).

**Figure 3 cancers-15-00247-f003:**
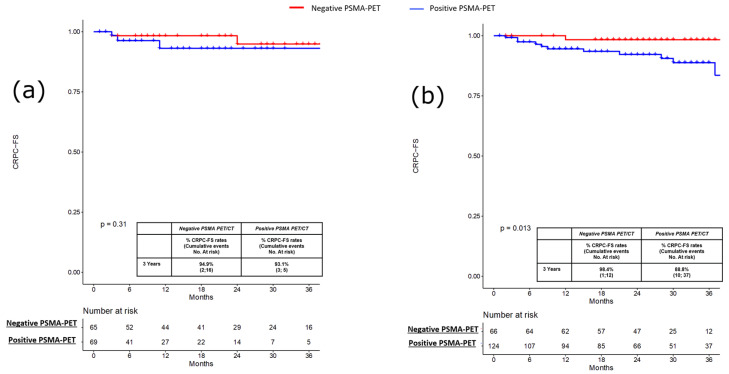
(**a**) Kaplan-Meier curve depicting Castration Resistant Prostate Cancer Free Survival (CRPC-FS) rates in the pre-salvage setting (*n* = 134) according to PSMA-PET results (namely, positive vs. negative); (**b**) Kaplan-Meier curve depicting Castration Resistant Prostate Cancer Free Survival (CRPC-FS) rates in the post-salvage setting (*n* = 190) according to PSMA-PET results (namely, positive vs. negative).

**Table 1 cancers-15-00247-t001:** Descriptive characteristics in the overall population (*n* = 324), in the pre-salvage (*n* = 134) and post-salvage (*n* = 190) setting stratifying according to PSMA-PET results (namely, positive vs. negative).

Overall		Pre-Salvage Setting, *n* = 134 (41.4%)			Post-Salvage Setting, *n* = 190 (58.6%)		
		Positive PSMA-PET	Negative PSMA-PET	*p* Value	Positive PSMA-PET	Negative PSMA-PET	*p* Value
Patients, *n* (%)	324 (100)	69 (52)	65 (48)	-	124 (65)	66 (35)	-
Age Median (IQR)	63 (57–68)	64 (59–70)	63 (56–68)	0.4	62 (56–66)	65 (59–70)	0.02
PSA level at RP (ng/mL) Median (IQR)	8.34 (5.51–12.88)	7.45 (6–13.22)	8.54 (4.33–12)	0.4	7.34 (2.33–13.85)	7.07 (4.85–9.97)	0.8
pT stage, *n* (%) pT2 pT3a pT3b-pT4	102 (31.5) 108 (33.3) 114 (35.2)	12 (17.4) 23 (33.3) 34 (49.3)	31 (47.7) 17 (26.2) 17 (26.2)	0.01	40 (32.3) 41 (33.1) 43 (34.7)	19 (28.8) 27 (40.9) 20 (30.3)	0.6
pN stage, *n* (%) pNx pN0 pN1	69 (21.3) 178 (54.9) 77 (23.8)	9 (13) 34 (49.3) 26 (37.7)	15 (23.1) 38 (58.5) 12 (18.5)	0.03	30 (24.2) 64 (51.6) 30 (24.2)	15 (22.7) 42 (63.6) 9 (13.6)	0.2
Pathologic ISUP group, *n* (%) ISUP 1–3 ISUP 4–5	157 (48) 167 (52)	24 (34.8) 45 (65.2)	45 (69.2) 20 (30.8)	≤0.001	58 (46.8) 66 (53.2)	30 (45.5) 36 (54.5)	0.9
Adjuvant Radiotherapy, *n* (%) Yes No	88 (27.2) 236 (72.8)	50 (72.5) 19 (7.5)	51 (78.5) 14 (21.5)	0.4	38 (30.6) 86 (69.4)	17 (25.8) 49 (4.2)	0.5
PSA level at PET PSMA, ng/mL Median (IQR)	0.5 (0.28–1.2)	0.8 (0.31–1.99)	0.33 (0.25–0.56)	≤0.001	0.66 (0.31–1.45)	0.48 (0.28–1)	0.06

PSMA: Prostate Specific Membrane Antigen; PET: Positron Emission Tomography; IQR: interquartile range; PSA: prostate specific antigen; RP: radical prostatectomy; ISUP: International Society of Urological Pathology.

**Table 2 cancers-15-00247-t002:** Oncologic outcomes after PSMA-PET results in the overall population (*n* = 324), in the pre-salvage (*n* = 134) and post-salvage (*n* = 190) setting stratifying according to PSMA-PET results (namely, positive vs. negative).

Overall		Pre-Salvage Setting, *n* = 134 (41.4%)			Post-Salvage Setting, *n* = 190 (58.6%)		
		Positive PSMA-PET	Negative PSMA-PET	*p* Value	Positive PSMA-PET	Negative PSMA-PET	*p* Value
Patients, *n* (%)	324 (100)	69 (51)	65 (49)	-	124	66	-
Treatment performed after PSMA-PET, *n* (%) Prostate bed RT/whole pelvis RT sLND Lymph node SRBT Bone SRBT sLND + SBRT Pelvic RT + bone SBRT Cyberknife Cryotherapy Metastases resection ADT ADT + ARTA/Chemo Observation	88 (27.2) 45 (13.9) 9 (2.8) 18 (5.6) 6 (1.9) 19 (5.9) 2 (0.6) 4 (1.2) 1 (0.3) 69 (21.3) 4 (1.2) 59 (18.2)	18 (26.1) 7 (10.1) 1 (1.4) 4 (5.8) 3 (4.3) 17 (24.6) 0 (0) 0 (0) 0 (0) 8 (11.6) 1 (1.4) 10 (14.5)	26 (40) 1 (1.5) 0 (0) 0 (0) 0 (0) 0 (0) 0 (0) 0 (0) 0 (0) 14 (21.5) 0 (0) 24 (36.9)	≤0.001	26 (21) 37 (29.8) 8 (6.5) 14 (11.3) 3 (2.4) 2 (1.6) 2 (1.6) 4 (3.2) 1 (0.8) 15 (12.1) 3 (2.4) 9 (7.3)	18 (27.3) 0 (0) 0 (0) 0 (0) 0 (0) 0 (0) 0 (0) 0 (0) 0 (0) 32 (48.5) 0 (0) 16 (24.2)	≤0.001
PSA at recurrence after PSMA-PET, ng/mL Median (IQR)	0.85 (0.43–1.95)	0.85 (0.23–1.85	0.58 (0.17–2.9)	0.9	0.98 (0.5–2.54)	1.17 (0.48–1.88)	0.5
Time to PSA recurrence, months Median (IQR)	9 (3–23)	11.5 (5–24)	6 (3–25)	0.9	8 (3–17)	14 (4–24)	0.3
Metastatic recurrence, *n* (%) Yes No	47 (14.5) 277 (85.5)	4 (5.8) 65 (94.2)	3 (4.6) 62 (95.4)	0.8	35 (28.2) 89 (71.8)	5 (7.6) 61 (92.4)	0.001
Time to metastatic recurrence, months Median (IQR)	21 (8–32)	11 (3–32)	22 (8–35)	≤0.001	17 (8–25)	24 (11–34)	0.003
CRPC, *n* (%) Yes No	26 (8) 298 (92)	4 (5.8) 65 (94.2)	3 (4.6) 62 (95.4)	0.8	18 (14.5) 106 (85.5)	1 (1.5) 65 (85.5)	0.004
Time to CRPC, months Median (IQR)	23 (9–33)	7.5 (3–37)	22 (8–36)	0.001	29 (8–40)	27 (26–33)	0.3
Overall Mortality, *n* (%) Yes No	9 (2.8) 315 (97.2)	0 69 (100)	1 (1.5) 64 (98.5)	0.3	7 (95.6) 117 (94.4)	1 (1.5) 65 (85.5)	0.2
Cancer specific mortality, *n* (%) Yes No	3 (0.9) 321 (99.1)	0 69 (100)	0 65 (100)	-	3 (2.4) 121 (97.6)	0 66 (100)	0.2
Follow up (months) from PSMA-PET Median (IQR)	23 (10–34)	8 (3–21)	23 (9–37)	≤0.001	27 (15–40)	27 (23–33)	0.6
Follow up (months) from RP Median (IQR)	62 (30–108)	35 (16–74)	63 (45–100)	≤0.001	68 (0–118)	74 (50–141)	0.02

sLND: salvage lymph node dissection; SBRT: stereotactic body radiotherapy; ADT: androgen deprivation therapy; ARTA: androgen receptor targeted agents; PSA: prostate specific antigen; BCR: biochemical recurrence; CRPC: castration resistant prostate cancer.

**Table 3 cancers-15-00247-t003:** Multivariate Cox regression to predict Progression free survival and Metastasis free survival in the overall population (*n* = 324).

Variables	Progression		Metastasis	
	HR (95% C.I.)	*p* Value	HR (95% C.I.)	*p* Value
Age (years)	0.97 (0.95–0.99)	0.04	-	-
Clinical setting for PSMA-PET Pre-salvage setting Post-salvage setting	1.0 (Ref) 1.31 (0.86–1.98)	0.2	1.0 (Ref) 1.40 (0.93–2.10)	0.1
Pathologic stage pT2 pT3a pT3b-pT4	1.0 (Ref) 1.47 (0.90–2.39) 1.84 (1.10–3.09)	0.1 0.02	1.0 (Ref) 1.49 (0.93–2.41) 2.03 (1.26–3.26)	0.09 0.003
Pathologic ISUP group ISUP 1– 3ISUP 4–5	1.0 (Ref) 0.92 (0.60–1.46)	0.7	1.0 (Ref) 0.87 (0.93–1.28)	0.5
pN stage, *n* (%) pNx/pN0 pN1	(Ref) 1.04 (0.66–1.65)	0.8	-	-
ADT at salvage treatment	0.58 (0.40–0.85)	0.005	0.54 (0.37–0.78)	0.001
PSA at PSMA-PET (ng/mL)	0.99 (0.97–1-02)	0.4	-	-
PSMA-PET result Negative Positive	1.0 (Ref) 2.15 (1.42–3.25)	<0.001	1.0 (Ref) 2.37 (1.60–3.50)	<0.001

HR: Hazzard ratio

## Data Availability

Not applicable.

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
