# Peer review of "The Impact of PSMA-PET on Oncologic Control in Prostate Cancer Patients Who Experienced PSA Persistence or Recurrence"

_cancers, 2022, doi:10.3390/cancers15010247_

Round 1
Reviewer 1 Report
The authors provide an informative article about PSMA-PET as a prognostic tool in BCR patients after radical prostatectomy. This study is an important contribution to topic, methods and materials are well described and conclusions appropriately supported by the data.
Comments:
Have PSMA-PET positive lesions had to be confirmed in conventional imaging before MDT or was treatment solely based on PSMA-PET?
Was there a cutoff level for the SUV in PSMA PET?
In case of PSA positive M1 situation, did patients also receive local irradiation of the prostate bed even in the absence of local recurrence?
The manuscript suffers from several grammar / spelling errors that should be corrected.
Author Response
We would like to thank the Reviewer for his/her comment and for the good opinion about our researches on this topic.
Have PSMA-PET positive lesions had to be confirmed in conventional imaging before MDT or was treatment solely based on PSMA-PET?
We would like to thank the Reviewer for his/her comment. MDT targeted to PSMA-PET positive lesions were administered with no further conventional imaging. In some case of suspicious lesion at PSMA-PET, histologic confirmation was achieved (in a minority of patients) or confirmatory further examinations (conventional imaging or further PET investigation with PSMA or choline). However, in the vast majority of cases MDT was based only on PSMA. Manuscript was modified as follows:
-Page 3, lines 134-136: “In vast majority of cases, MDT was targeted to PSMA-PET positive lesions with no further confirmation by conventional imaging.”
Was there a cutoff level for the SUV in PSMA PET?
We would like to thank the Reviewer for his/her comment. A cut-off level for the SUV in PSMA-PET was not used since it’s not recommended by guidelines. As reported in methods section all PSMA-PET images were locally reviewed prior data sharing, independently by two experienced nuclear medicine physicians and according to reporting international guidelines (doi:10.1007/s00259-021-05245-y, doi:10.1007/s00259-017-3670-z, doi:10.1007/s00259-017-3725-1)
In case of PSA positive M1 situation, did patients also receive local irradiation of the prostate bed even in the absence of local recurrence?
We would like to thank the Reviewer for his/her comment. In case of M1 with no focal uptake at PSMA-PET in the prostate bed, patients received only treatment of metastases with no prostate bed irradiation, besides high clinic risk of concomitant local relapse, considering previous pelvic RT, ISUP 4-5, pT3 and R status. In this case, a concomitant RT in the prostate bed was considered
The manuscript suffers from several grammar / spelling errors that should be corrected.
We would like to thank the Reviewer for his/her comment. English revision of the manuscript was performed.
Reviewer 2 Report
This manuscript concerning PSMA-Pet imaging in patients with BCR after prostatectomy is very timely. Many manuscript are being composed (up to now in 2022 >580 manuscripts on PubMed) and published. That means that new manuscripts must be above average to say the least and for me this manuscript doesn’t make the cut.
The thought behind the manuscripts is clear: “evaluate the oncologic outcomes of patients who received PSMA-Pet to stage the disease during PCa recurrence”. For this the authors made a retrospective analysis of current data. I have some concerns.
First (major): continuous throughout the manuscript the pre-salvage and post-salvage patients are discussed but it is very confusing. Pre-salvage and post-salvage are 2 very different stages of prostate cancer and can not be compared in the manner used in this manuscript. I would propose 2 different sub-sets in the manuscript (first concentrate on the pre-salvage setting, and later discuss the results of the post-salvage setting). One could even argue to produce 2 different manuscripts altogether.
Second (major): the preferred PSMA-PET tracer was 68Ga-PSMA-11. This tracer has been extensively studied, als in the BCR-setting post-prostatectomy. The uptake of the 68Ga in the urine makes it a less accurate tracer for local recurrence. That been said, the negative PSMA scans (namely pre-salvage) could still harbor a local recurrence. What was the policy for patients with a BCR and negative PSMA-scan? This also because of the guideline advise to use salvage radiation post-prostatectomy with a PSA>0.2 and no abnormality on (PSMA-) imaging.
Third (minor): in table 1 the median PSA-level of the pre-salvage patients with a negative PSMA was considerable lower than the positive PSMA patients. Please elaborate: this could be a bias concerning the negative scans.
Last (minor): the OLV-hospital Aalst is a non-academic hospital. Please correct
Author Response
-First (major): continuous throughout the manuscript the pre-salvage and post-salvage patients are discussed but it is very confusing. Pre-salvage and post-salvage are 2 very different stages of prostate cancer and can not be compared in the manner used in this manuscript. I would propose 2 different sub-sets in the manuscript (first concentrate on the pre-salvage setting, and later discuss the results of the post-salvage setting). One could even argue to produce 2 different manuscripts altogether.
We would like to thank the Reviewer for his/her comment. We agree that pre-salvage and post-salvage setting are different populations and our goal was not to compare the two settings, but to show the impact of PSMA-PET results (positive vs negative) in the two different populations to clarify the different benefit that could have in both populations. As consequence we modified the results section discussing separately the two settings, firstly the pre-salvage and secondly the post-salvage setting separately. Please see the modified results section as follows:
Page 7, lines 201-223:
3.1 Pre-salvage setting (n=134)
In the pre-salvage setting, PSMA-PET positivity rate was 52% (69 out of 134 patients). Patients with positive PSMA-PET had significant higher pathologic T stage, pN1 status, pathologic ISUP group and median PSA level at PSMA-PET compared to patients with negative ones (all p≤0.03; Table 1). Supplementary Table 1 shows baseline characteristics in patients with positive PSMA-PET (n=193) stratifies according to localization of positive lesions (local and/or N1 vs M1). Supplementary Table 2 shows baseline characteristics in patients with positive PSMA-PET (n=193) stratifies according to the stage of disease (oligorecurrent vs. polirecurrent). PFS, MFS and CRPC-FS estimates at 3 years were 66.2% vs. 38.9% (p=0.3; Figure 1a), 95.2% vs 73.7% (p=0.2; Figure 2a) and 94.9% vs 93.1% (p=0.3; Figure 3a) in patients with negative vs. positive PSMA-PET, respectively.
3.2 Pre-salvage setting (n=190)
In the post-salvage setting, PSMA-PET positivity rate was 65% (124 out of 190 patients). Patients with positive PSMA-PET had significant lower age (p=0.02), but no significant difference were found concerning pathologic characteristics compared to patients with negative ones (Table 1). PFS, MFS and CRPC-FS estimates at 3 years were 59.5% vs. 29.1% (p<0.001; Figure 1b), 92.7% vs. 65.1% (p<0.0004; Figure 2b) and 98.8% vs 88.8% (p=0.01; Figure 3b) in patients with negative vs. positive PSMA-PET, respectively.
-Second (major): the preferred PSMA-PET tracer was 68Ga-PSMA-11. This tracer has been extensively studied, als in the BCR-setting post-prostatectomy. The uptake of the 68Ga in the urine makes it a less accurate tracer for local recurrence. That been said, the negative PSMA scans (namely pre-salvage) could still harbor a local recurrence. What was the policy for patients with a BCR and negative PSMA-scan? This also because of the guideline advise to use salvage radiation post-prostatectomy with a PSA>0.2 and no abnormality on (PSMA-) imaging. negative scans.
We would like to thank the Reviewer for his/her comment. We agree that PSMA-PET may not be diagnostic for local recurrent disease. So, in case of PCa patients with BCR and no significant uptake at PSMA-PET (negative scan), the primary indication was to perform sRT. As reported in the methods section (Page 3, lines 139-143): “Patients with negative PSMA-PET were treated with best clinical practice (including salvage prostate bed RT/whole pelvis RT or systemic therapies or observation), according to patients’ clinical status (including eventual comorbidities), previous treatments and multidisciplinary tumor board decisions”
-Third (minor): in table 1 the median PSA-level of the pre-salvage patients with a negative PSMA was considerable lower than the positive PSMA patients. Please elaborate: this could be a bias concerning the negative scans.
We would like to thank the Reviewer for his/her comment. We agree with your comment, so the discussion was modified as follows:
Page 8, lines 269-271: “Moreover, low PSA level may reflect a negative PSMA-PET and could be a potential bias for PSMA-PET diagnostic performance”.
Last (minor): the OLV-hospital Aalst is a non-academic hospital. Please correct
We would like to thank the Reviewer for his/her comment. We modified academic with tertiary centers as follows:
-Page 2, lines 92-94: “The cohort of patients included in this study was enrolled through an open label, multicenter, retrospective analysis in three tertiary academics high-volume European centers”